# The basic psychological needs satisfaction and frustration scale at work: A validation in the Polish language

**Michał Szulawski**[1]*, **Łukasz Baka**[1], **Monika Prusik**[2], **Anja H. Olafsen**[3]

**1** The Maria Grzegorzewska University, Warszawa, Poland, **2** University of Warsaw, Warsaw, Poland, **3** University of South-Eastern Norway, Hønefoss, Norway

* mszulawski@aps.edu.pl

**Data Availability Statement:** All relevant data are within the manuscript and its Supporting Information files.

## Abstract

The aim of this research project was to validate the work-related version of the Basic Psychological Need Satisfaction and Frustration Scale (BPNSFS) into the Polish language and culture. Although studies have demonstrated the benefits associated with basic psychological need satisfaction and the costs associated with need frustration at work, the concept of needs has been neglected both in Polish scientific research and in practical organizational studies. The adaptation of the BPNSFS-Work Domain may change this situation and stimulate research in the Polish community. The scale has been validated in a sample consisting of three occupational groups: healthcare workers, education staff and customer service workers ($N$ = 1315, $M_{age}$ = 43.8). The findings suggest that the Polish scale has robust psychometric features. The CFA analysis proves that the scale has a six-dimensional structure similar to the original scale. These dimensions show satisfactory to high Cronbach's $\alpha$ and McDonalds $\omega$ reliability, and high criterion validity is shown by association of the six need dimensions with correlates of both positive (i.e., engagement, job crafting and self-efficacy) and negative aspects of work (i.e., burnout and stress). The structure of the scale is the same in all three occupational groups, although the regression weights and covariances are only partially invariant. The validated version of the BPNSFS-Work Domain can be used in future basic and applied studies in the paradigm of self-determination theory.

## Introduction

The issue of how to build engagement and motivation and at the same time avoid stress and burnout is an important topic within many organizations [1–3]. Studies suggest that low-quality motivation may contribute towards decreased well-being, higher stress levels and burnout [4–7]. On the other hand, high-quality motivation is associated with high performance [8, 9] and job satisfaction [10]. Research indicates that, in a professional environment, both the occurrence of positive phenomena, such as high-quality motivation, performance, job satisfaction and well-being, as well as a lack of negative phenomena, such as stress, burnout and disengagement are related to the concept of basic psychological needs (basic psychological needs

**Funding:** M.S. BSTP 40/20-I The Polish Ministry of Eduation https://www.gov.pl/web/edukacja-i-nauka. The funders had no role in study design, data collection and analysis, decision to publish, or preparation of the manuscript.

**Competing interests:** The authors have declared that no competing interests exist.

theory, or BPNT). BPNT is a subtheory of self-determination theory (SDT) [11–13], and depicts people as having three needs—autonomy, relatedness and competence—which are essential for growth, optimal motivation and well-being. The needs are universal, meaning that they play an important role across cultures and environments [14, 15]. In professional environments, autonomy satisfaction occurs when at work one experiences psychological freedom, creation and the possibility to choose, while autonomy frustration represents a feeling of being controlled and pressured. Competence satisfaction involves feeling effective and capable at work, while competence frustration denotes a feeling of inadequacy and failure at the tasks one is responsible for. Relatedness satisfaction involves the sense of being warmly connected to people from one's work environment (e.g., coworkers, clients), while relatedness frustration denotes feelings of loneliness, ostracism or rejection [12, 16, 17]. According to SDT [12] there are six different types of motivational regulation, which vary in their antecedents, the degree of perceived autonomy, and effects on behavior. The six types of motivation are amotivation, external regulation, introjection, identification, integration and intrinsic motivation. The differentiation between low (poor) or high (good) quality motivations is more general (than in OIT theory) which assumes that all controlled types of motivation (external regulation, introjection) are considered low (poor) quality motivation and all autonomic types of motivation (identification, integration and intrinsic motivation) are considered high (good) quality motivation. The three basic psychological needs according to the SDT theory serve as psychological nutrients for developing high quality motivation [12].

At first, research within BPNT was focused on the phenomena of basic psychological needs satisfaction; with time, however, increasing evidence has shown that low need satisfaction does not simply mean the presence of need frustration [13, 18] and, as a consequence, the study of need frustration received increased attention. The difference between low satisfaction and frustration may be illustrated by the example of competence need, which may not be satisfied if a person on a given day has difficulties in finishing some planned task or project; yet, he or she still may not feel incompetent or failure in what they are working on, which would signal the presence of need frustration [13]. In general, a need is frustrated when a more active threat to it takes place.

Within organizational research, the satisfaction of basic psychological needs has proven to be associated with a variety of different performance-related and well-being-related outcomes, including self-reported and manager-reported performance [19, 20], knowledge sharing [21], organizational citizenship behavior [22], psychological adjustment [19], vitality [23] and proactive attitude [24]. Studying the separate dynamics of need satisfaction and need frustration appeared fruitful because need frustration was found to be especially predictive for the 'dark' side of individuals' professional functioning. That is, need frustration robustly relates to stress at work [20], ill-being [25], disengagement [26] and poor sleep [27].

Studies around the world occupied with the concepts of need satisfaction and frustration typically use the Basic Psychological Need Satisfaction and Frustration Scale (BPNSFS) [14], a popular measurement tool that includes a collection of items from satisfaction and frustration subscales. In particular, BPNSFS comprises 24 items and six subscales, namely, autonomy satisfaction and frustration, competence satisfaction and frustration, and relatedness satisfaction and frustration; each of the subscales consists of four items. The first validation studies concerning the scale were conducted in four countries (Belgium, China, Peru and the US) and confirmed its psychometric characteristics. The validation studies showed that the reliability of the six dimensions was satisfying in each of the four cultures. Moreover, the findings of well-being and ill-being were invariant across the four studied cultures [14], suggesting that the role of the needs is universal. Since the original publication, the results have been replicated in numerous countries, with various translations and adaptations to various life domains [4]. The

general version of the scale has been validated in, among others, Japanese [28], Portuguese [29], Hebrew [30] Polish [31]and Arabic [32]. Then, the need of studies in the work environment made researchers prepare a scale for this purpose [33]. The first validation of the BPNSFS at work was conducted in Norwegian and English [33, 34].

## Present research

Although numerous studies have demonstrated the benefits associated with need satisfaction and the costs associated with need frustration at work, the role of basic psychological needs in the Polish work environment and scientific research has been neglected [35]. With the BPNSFS-Work Domain scale, Polish researchers will be able to join the worldwide community conducting studies within the self-determination theory framework; these results may have important implications for Polish work life. As a consequence, the scale will serve not only scientific but also practical purposes. It is important to be able to measure the construct of basic psychological needs that have proven to be central for a number of beneficial organizational factors. Studies concerning the basic psychological needs may be useful to managers and HR departments in organizations for readjusting or changing management strategies. For these reasons, the purpose of this project was to validate the BSNSFS-Work Domain at work into Polish language and culture. It is worth noting that the general version of the BSNSFS has also been adapted into Polish language, however the present version is intended to verify the basic psychological needs inclusively in the work context, in contrast to the general version of the scale which is intended to measure the basic needs in general population [31].

The validation project consisted of several steps. First, we conducted a full language adaptation using a parallel blind technique and blind back-translation to check accuracy [36]. Second, we conducted a study to examine the dimensional structure of the Polish scale and to examine invariance between the three research groups involved. Third, we investigated associations of the scale with theoretically-related constructs relevant to the work environment (i.e., work engagement, job crafting, self-efficacy, burnout and stress). We compared the construct of needs with other, important for work environment constructs, in contrast to the general version of the scale where the associations were checked with negative mood and general life satisfaction [31]. In this research, we anticipated the scale to have the same or similar factorial structure as the original and expected moderate associations between the satisfaction and frustration of the basic psychological needs at work and self-efficacy, work engagement, job crafting, stress, exhaustion, demonstrating their relatedness but not interchangeability. More specifically, we expected that satisfaction of the needs would be positively related to self-efficacy, work engagement and job crafting, while frustration of the needs would be negatively correlated with these constructs. We anticipated that satisfaction of the basic psychological needs would be negatively related to stress and exhaustion, and frustration of the needs would be positively related to them.

## Methods

### Procedure and participants

In order to adapt BPNSFS-Work Domain [14, 33] into Polish, the items of the original scale were translated into Polish by a bilingual translator; another bilingual person then back-translated the scale. Next, we consulted the back-translated English version of the scale with the authors of the original, universal version of the scale and whenever inconsistencies were found, we continued the back-translation procedure until a consensus was reached that the translation was the best representation of the meaning of the items in Polish. Then, we collected data on the Polish version in order to test its factorial structure and invariance of structure between the three researched groups, as well as calculating criterion validity.

The respondents of the study ($N$ = 1315) included employees from three professional sectors that work in direct contact with other people (e.g., patients, students or clients): (1) healthcare workers ($n$ = 440; e.g., doctors, nurses, midwives, paramedics, rehabilitation specialists, physiotherapists, medical assistants), (2) education staff ($n$ = 439; e.g., school teachers, academic teachers, kindergarten teachers, librarians, school principals) and (3) customer service workers ($n$ = 436; e.g., government officials, office workers, personnel of human resources departments, employees of customer service departments, bank employees, hoteliers, receptionists). The study was conducted between May and September 2020 in 97 facilities (e.g., hospitals, schools, offices) where the respondents were employed. All participants were treated in accordance with the ethical guidelines of the Helsinki Declaration and received a hard copy of the questionnaires along with a letter explaining the purpose of the study. Full confidentiality of data and anonymity were secured. Participants were asked to fill out the questionnaires and seal them in envelopes, which were subsequently collected by research assistants. Out of 2,000 distributed questionnaires, 1542 (77%) were returned and 1315 (57% of the original pool) were at least 75% complete and subsequently used for the data analysis. The analyzed group consisted of 945 women (72%) and 370 men (28%) between 20 and 71 years of age ($M$ = 43.8, $SD$ = 11.13). Work experience ranged from 1 to 50 years ($M$ = 19.50, $SD$ = 10.99).

## Measures

**Basic psychological needs.** Autonomy, relatedness, and competence satisfaction and frustration were assessed by the linguistically-adapted experimental version of BPNSFS-Work Domain. The scale consists of 24 items, four items for each of the six subscales (i.e., *autonomy satisfaction*, *autonomy frustration*, *relatedness satisfaction*, *relatedness frustration*, *competence satisfaction and competence frustration*). Respondents answered the questions concerning their feelings about their jobs during the previous four weeks, on a 7-point response scale ranging from 1 (*strongly disagree*) to 7 (*strongly agree*). The reliability coefficients are presented in the results section of this paper.

**Stress.** Stress was measured with a subscale of the Polish version of the Copenhagen Psychosocial Questionnaire (COPSOQ II) [37, 38], referring to the frequency of experiencing various unpleasant mental states, such as irritation, tension or relaxation problems during the previous four weeks. The subscale includes four questions (e.g., *How often have you been tense*?) assessed on a 5-point response scale ranging from 1 (*all the time*) to 5 (*not at all*). The scale showed decent statistical parameters, for a short scale, concerning Cronbach's alpha ($\alpha$ = .62) and McDonald's omega ($\omega$ = .63).

**Emotional exhaustion.** Emotional exhaustion was measured with the subscale of the Polish version of the Oldenburg Burnout Inventory (OLBI) [39, 40]. The subscale consists of eight items (e.g., *After my work, I regularly feel worn out and weary*) assessed on a 4-point response scale ranging from 1 (*I completely agree*) to 4 (*I completely disagree*). Recoding is required for half of the questions on the subscale. Good internal reliability coefficients for the scale were obtained with $\alpha$ = .79 and $\omega$ = .79.

**Occupational self-efficacy.** Self-efficacy was measured with a short form of the Polish version of the Occupational Self-Efficacy Scale (OSS-SF); [41, 42]. The OSS-SF is composed of six items that concern self-efficacy beliefs related to demands and problems in the workplace (e.g., *When I am confronted with a problem in my job, I can usually find several solutions*), assessed on a 6-point response scale, ranging from 1 (*strongly disagree*) to 6 (*strongly agree*). In the present study, Cronbach's alpha and McDonald's omega coefficients for the OSS-SF were: $\alpha$ = .92 and $\omega$ = .92, respectively.

**Job crafting.** To assess this variable, the Polish version of the Job Crafting Scale (JCS); [43, 44] was used. This 16-item tool refers to the Job Demands-Resources model [45] and consists

of four subscales: (1) crafting of structural job resources related to skills, learning and professional development (five items, e.g., *I try to learn new things at work*), (2) crafting of social job resources related to quality of interaction with supervisors and colleagues (five items, e.g., *I ask my supervisor to coach me*), (3) crafting of challenging job demands related to proactivity in the development of new and interesting job demands (five items, e.g., *When an interesting project comes along, I offer myself proactively as a project co-worker*) and (4) crafting of hindrance job demands related to avoiding or minimizing cognitively or emotionally onerous demands (six items, e.g., *I organize my work so as to minimize contact with people whose expectations are unrealistic*). The responses are given on a 5-point scale ranging from 1 (*never*) to 5 (*very often*). A total index of job crafting was used in the study ($\alpha$ = .89 and $\omega$ = .89).

**Work engagement.** This variable was assessed with the Polish version of the Utrecht Work Engagement Scale (UWES), [46, 47]. It is a 17-item instrument measuring three components of work engagement, including vigor (e.g., *At my job, I feel strong and vigorous*), dedication (e.g., *My job inspires me*) and absorption (e.g., *I feel happy when I am working intensely*). All items are scored from 0 (*never*) to 6 (*every day*). An aggregated index of work engagement was used in the present study, where a high global score means strong work engagement. The reliability coefficients for the UWES were $\alpha$ = .95 and and $\omega$ = .95, respectively.

## Data analyses

SPSS version 26 was used to compute descriptive statistics, reliability (by means of Cronbach's $\alpha$ and McDonald's $\Omega$) and correlation analyses. To check if the structure of the Polish BPNSFS-Work Domain is supported by the empirical data, we conducted confirmatory factor analysis (CFA) using AMOS version 26. A CFA offers a more rigorous test than an exploratory factor analysis and is more suitable for testing the structure of psychological tools where previous research is already available and there are clear assumptions regarding the structure of the latent constructs (number and composition). As multivariate normality is assumed for most CFA estimation methods, and departures from multivariate normality can have a significant impact on CFA estimations [48], descriptive analytical measures were calculated prior to conducting the CFA. We also checked other assumptions, such as linearity between items (on random pairs of items), analyzed occurrence of outlying cases and examined the pattern of missing data. This resulted in elimination of six cases (*N* = 1309 for the remaining sample) from CFA analysis due to, for example, zero variance on items included in the scale.

Following BPNT [16], three different CFA models were explored and compared: (a) a 6-factorial model corresponding to the BPNSFS, which consists of six separate components (three for satisfaction and three for frustration of needs), (b) a 3-factorial model capturing the satisfaction and frustration of the three needs and, (3) a 1-factor model aggregating the six dimensions studied into one factor. Based on Hoyle's [49] recommendations, and according to the multi-faceted approach to the assessment of model fit [50], the following goodness-of-fit indices were considered: the root mean square error of approximation (RMSEA), the chi-square to *df* ratio (CMIN/df), the goodness of fit-index (GFI), the adjusted goodness-of-fit index (AGFI), the comparative-fit index (CFI) and the HOELTER-fit index. RMSEA values lower than 0.05 are considered good, while values lower than 0.08 are considered acceptable. CMIN/df values lower than 2.00 are usually considered very good, while values from 2 to 5 indicate acceptable model fit. GFI, AGFI and CFI values equal to or higher than 0.90 are considered acceptable, while values equal to or higher than 0.95 are considered good [48]. HOELTER values higher than 200 indicate good model fit. The $\chi 2$ values are provided for each analysis but are not used to evaluate the overall model fit, as the $\chi 2$ test is overly sensitive for large samples [48]. As our participants in the study were from three occupational niches, we also conducted

**Table 1. Means and standard deviations of basic psychological needs, self-efficacy, engagement, job crafting, stress, exhaustion and age.**

| Variable | Descriptive Statistics | | | |
| --- | --- | --- | --- | --- |
| | M | SD | skewness | kurtosis |
| Stress | 269.78 | 94.92 | -0.99 | 0.82 |
| Exhaustion | 17.79 | 4.18 | -0.23 | -0.04 |
| Self-Efficacy | 28.51 | 5.13 | -0.64 | 0.40 |
| Engagement | 72.01 | 15.60 | -0.55 | 0.44 |
| Job Crafting | 71.77 | 11.38 | 0.12 | -0.10 |
| Autonomy Support | 19.75 | 4.19 | -0.25 | -0.19 |
| Autonomy Frustration | 14.97 | 4.44 | -0.05 | -0.15 |
| Relatedness Support | 20.65 | 4.36 | -0.41 | 0.02 |
| Relatedness Frustration | 10.86 | 5.34 | 0.50 | -0.72 |
| Competence Support | 21.92 | 4.57 | -0.67 | 0.08 |
| Competence Frustration | 10.17 | 5.43 | 0.67 | -0.48 |
| Age | 43.8 | 11.1 | 0.03 | -0.96 |

Note. N = 1315. Means for the indices based on sum were presented because particular scales had different ranges.

invariance analysis including different types of invariance testing (e.g., configural and metric) following steps recommended by the Vandenberg and Lance approach [51] and summarized by Milfont and Fischer [52].

### Ethics statement

The research was approved by the Maria Grzegorzewska Ethics Committee. The approval was obtained in the form of the written consent (no. BSTP 40/20-I).

## Results

### Preliminary analyses

The preliminary analyses of data started by computing the descriptive statistics for the study variables. Table 1 includes means, standard deviations, skewness and kurtosis.

In the next step we conducted the reliability of the scales. The Cronbach's $\alpha$ and McDonalds $\omega$ scores are presented in Table 2.

### CFA including invariance analyses

The analytical work related to CFA analysis consisted of three general objectives. Firstly, we wanted to decide which model fits the data better: the original six-factor model against a one-factor model, or a model consisting of three factors as postulated by some authors. Secondly, we wanted to investigate some potential avenues for improvement of the chosen model by constraining or freeing certain parameters based on the results, by adding model re-

**Table 2. Cronbach's α and McDonalds ω of the six dimensions of Polish BPNSFS-work domain.**

| | Autonomy satisfaction | Relatedness satisfaction | Competence satisfaction | Autonomy frustration | Relatedness frustration | Competence frustration |
| --- | --- | --- | --- | --- | --- | --- |
| α Cronbach | .68 | .81 | .79 | .64 | .82 | .82 |
| ω McDonalds | .69 | .81 | .80 | .66 | .82 | .82 |

specifications based on error covariances or by proposing alternative structures of the model. Thirdly, we wanted to test whether our results related to the chosen model are invariant, especially for measurement weights and structural covariances, across the three occupational groups within the sample.

Therefore, first we conducted a confirmatory factor analysis to verify if the six-dimensional structure of the Polish BPNSFS-Work Domain has support in the data (Table 4). Results showed that the six-factor model yielded a relatively good overall fit, which was also a better fit in comparison to the three-factor model, $\Delta\chi^2(12) = 2729.01$, $p < .001$, and one-factor model, $\Delta\chi^2(16) = 3016.26$, $p < .001$. Both the three-factor and the one-factorial model had unacceptable model fit indices. The six-factor model not only had a better and an acceptable fit, but it was characterized by good regression weights, mostly exceeding the value of .50, except for items 1 and 18. The unsatisfactory CMIN/$df$ was expected due to the large sample size. As correlational coefficients between autonomy satisfaction and competence satisfaction, as well competence frustration and relatedness frustration exceeded .90, we decided to check if a four-factor model perhaps would better fit the data. The additional analysis showed that it not only did not have a better fit, but it actually had a significantly worse fit in comparison to the six-factor model ($\Delta\chi^2(8) = -327.51$, $p < .001$) (Table 3). We also checked whether adding some respecifications to the six-factor model would improve its fit. Indeed, adding modifications based on error covariances significantly improved the fit of the already acceptable six-factor model ($\Delta\chi^2(6) = 430.50$, $p < .001$). However, for the purity of further comparisons we decided to use the six-factor model with no modifications. Overall, analysis indicated that the six-factor model should be retained. Next, we proceeded with invariance analysis guided by the previously mentioned steps proposed e.g., by the Vandenberg and Lance approach [51] and summarized by Milfont and Fischer [52].

Three types of invariance were examined. First, we concentrated on configural invariance. This type of invariance was judged based on the fit indices, as well as size of regression weights for the overall baseline model (three groups combined) as well for the models for each of the three groups. Then, we examined metric invariance by analyzing differences between regression weights among the three examined groups. At the end, we also checked whether there were any differences in coefficients for covariances between retained factors among the three groups (structural covariance invariance).

As all the examined models—the overall baseline model with three groups combined and particular models for each group separately—were characterized by good or very good model fits and regression weights were mostly sufficiently high, we concluded that the data supported configural invariance (Table 4).

Based on the results of an omnibus metric invariance analysis for the six-factor baseline model, there was no metric invariance in the investigated three groups for measurement weights (baseline vs. measurement model) based on a $\chi^2$ difference test ($\Delta\chi^2 (36) = 67.33$, $p = .001$). However, with the $\chi^2$ difference test and larger sample sizes, even small differences become significant. Taking into account that the analysis met both the criteria proposed by Cheung & Rensvold [53] of ΔCFI not exceeding the difference of .01 as an indication of coefficients being invariant and additional criteria of ΔCFI < .01 coupled with ΔRMSEA < .015 and ΔSRMR < .030 [54, 55], the obtained results show that the weights could be treated as invariant, ΔCFI = .002.

Not the same could be concluded for measurement covariances, as both typically used criteria for structural covariance invariance ($\chi^2$ difference test and ΔCFI), which were not fulfilled for the comparison of measurement vs. structural model ($\Delta\chi^2 (90) = 263.21$, $p < .001$, ΔCFI = .015). As the invariance results for measurement weights were mostly but not entirely positive (based on ΔCFI and additional indices but not on a $\chi^2$ difference test) and results of invariance

**Table 4. Pairwise comparisons expressed in z-values for the measurement weights and structural covariances in multigroup six-factorial baseline model.**

| Measurement weights | $\beta_{G1}$ | $\beta_{G2}$ | $\beta_{G3}$ | $z_{G1-G2}$ | $z_{G1-G3}$ | $z_{G2-G3}$ |
|---|---|---|---|---|---|---|
| item no 19 > Autonomy satisfaction | .67*** | .58*** | .63*** | -1.94 | -1.69 | 0.24 |
| item no 13 > Autonomy satisfaction | .77*** | .75*** | .73*** | -1.76 | -1.40 | 0.35 |
| item no 7 > Autonomy satisfaction | .51*** | .56*** | .55*** | -1.24 | -1.10 | 0.11 |
| item no 1 > Autonomy satisfaction | .40*** | .55*** | .47*** | – | – | – |
| item no 18 > Autonomy frustration | .14* | .35*** | .26*** | 2.66** | 0.98 | -1.97 |
| item no 15 > Autonomy frustration | .61*** | .63*** | .49*** | 0.60 | -2.16* | -2.92** |
| item no 10 > Autonomy frustration | .71*** | .72*** | .78*** | -0.06 | -0.70 | -0.70 |
| item no 5 > Autonomy frustration | .58*** | .61*** | .65*** | – | – | – |
| item no 24 > Relatedness satisfaction | .70*** | .71*** | .65*** | -1.59 | -0.66 | 0.90 |
| item no 16 > Relatedness satisfaction | .74*** | .78*** | .76*** | -1.63 | -0.05 | 1.66 |
| item no 12 > Relatedness satisfaction | .80*** | .82*** | .84*** | -2.21* | 0.35 | 2.71** |
| item no 4 > Relatedness satisfaction | .56*** | .67*** | .58*** | – | – | – |
| item no 22 > Relatedness frustration | .66*** | .72*** | .67*** | 2.55* | 0.44 | -2.06* |
| item no 20 > Relatedness frustration | .75*** | .78*** | .75*** | 2.06* | 1.16 | -0.85 |
| item no 8 > Relatedness frustration | .74*** | .70*** | .78*** | 0.44 | 0.82 | 0.36 |
| item no 2 > Relatedness frustration | .73*** | .70*** | .69*** | – | – | – |
| item no 21 > Competence satisfaction | .69*** | .72*** | .69*** | -0.10 | -0.56 | -0.47 |
| item no 14 > Competence satisfaction | .78*** | .77*** | .74*** | -1.12 | -0.27 | 0.80 |
| item no 9 > Competence satisfaction | .68*** | .71*** | .77*** | -0.02 | 1.32 | 1.35 |
| item no 3 > Competence satisfaction | .63*** | .63*** | .60*** | – | – | – |
| item no 23 > Competence frustration | .79*** | .79*** | .78*** | 1.86 | 1.58 | -0.38 |
| item no 17 > Competence frustration | .69*** | .77*** | .76*** | 2.56* | 2.24* | -0.47 |
| item no 11 > Competence frustration | .77*** | .67*** | .79*** | 0.12 | 1.01 | 0.76 |
| item no 19 > Autonomy satisfaction | .72*** | .61*** | .65*** | – | – | – |
| **Structural covariances (standardized = Pearson's r)** | $r_{G1}$ | $r_{G2}$ | $r_{G3}$ | $z_{G1-G2}$ | $z_{G1-G3}$ | $z_{G2-G3}$ |
| Autonomy satisfaction>Autonomy frustration | -.24** | -.56*** | -.48*** | -3.72*** | -3.20** | 0.26 |
| Autonomy frustration>Relatedness satisfaction | -.34*** | -.47*** | -.40*** | -2.84** | -1.74 | 1.13 |
| Autonomy satisfaction>Relatedness satisfaction | .75*** | .87*** | .90*** | 3.84*** | 2.29* | -1.55 |
| Autonomy frustration>Relatedness frustration | .59*** | .83*** | .84*** | 1.97* | 3.26** | 1.46 |
| Relatedness satisfaction>Relatedness frustration | -.55*** | -.68*** | -.55*** | -2.91** | -0.76 | 2.12* |
| Autonomy satisfaction>Relatedness frustration | -.44*** | -.61*** | -.40*** | -2.36* | -0.59 | 1.66 |
| Autonomy frustration>Competence satisfaction | -.18** | -.45*** | -.44*** | -3.13** | -3.24** | -0.21 |
| Relatedness frustration>Competence satisfaction | -.63*** | -.61*** | -.58*** | -3.13** | 0.00 | 0.15 |
| Relatedness satisfaction>Competence satisfaction | .74*** | .84*** | .85*** | 3.34*** | 1.50 | -1.89 |
| Autonomy satisfaction>Competence satisfaction | .96*** | .99*** | .90*** | 2.08* | 0.82 | -1.24 |
| Competence frustration>Competence satisfaction | -.70*** | -.69*** | -.76*** | 0.17 | -0.43 | -0.58 |
| Competence frustration>Relatedness satisfaction | -.48*** | -.53*** | -.56*** | -1.72 | -1.11 | 0.65 |
| Competence frustration>Relatedness frustration | .93*** | .89*** | .91*** | -1.33 | 0.03 | 1.29 |
| Competence frustration>Autonomy frustration | .65*** | .74*** | .82*** | 0.51 | 2.26* | 1.79 |

Note.

\*\*\*$p < .001$

\*\*$p < .01$

\*$p < .05$; $r_{G1}$- $r_{G3}$. Pearson's r correlation coefficients for group 1, group 2 and group 3; $z_{G1-G2}$, $z_{G1-G3}$, $z_{G2-G3}$. Critical ratios expressed in z-values for comparisons between group 1, group 2 and group 3 for the coefficients of measurement weights, and for the structural covariances between latent variables.

**Table 3. Model fit indices for the examined models.**

| Models | $X^2$ | df | p | CMIN/DF | RMSEA [90% CI] | CFI | SRMR | N-HOELTER p < .05 | AIC | β range absolute |
|---|---|---|---|---|---|---|---|---|---|---|
| *First step–factorial structure* | | | | | | | | | | |
| Model 6-factor | 1409.45 | 237 | < .001 | 5.95 | .061 [.058, .065] | .91 | .05 | 255 | 1583.45 | [.26, .82] |
| Model 3-factor | 4138.46 | 249 | < .001 | 16.62 | .109 [.106, .112] | .72 | .09 | 91 | 4288.46 | [.11, .73] |
| Model 1-factor | 4425.71 | 253 | < .001 | 17.49 | .112 [.109, .115] | .70 | .09 | 87 | 4567.71 | [.09, .71] |
| Model 6-factor with re-specifications based on error covariances | 978.95 | 231 | < .001 | 4.24 | .050 [.047, .053] | .95 | .04 | 358 | 1164.95 | [.26, .83] |
| M4-factor model–with Autonomy and Competence satisfaction combined, and Competence and Relatedness frustration combined | 1736.96 | 245 | < .001 | 7.09 | .068 [.065, .071] | 89 | .05 | 213 | 1894.96 | [.02, .82] |
| *Second step—invariance* | | | | | | | | | | |
| M6-factor–GROUP 1 (G1) | 800.46 | 237 | < .001 | 3.38 | .074 [.068, .079] | .88 | .06 | 150 | 974.46 | [.14, .80] |
| M6-factor–GROUP 2 (G2) | 654.71 | 237 | < .001 | 2.76 | .064 [.058, .069] | .91 | .05 | 183 | 828.71 | [.35, .82] |
| M6-factor–GROUP 3 (G3) | 787.38 | 237 | < .001 | 3.32 | .073 [.068, .079] | .89 | .06 | 150 | 961.38 | [.26, .84] |
| M6-factor–Baseline model (Configural invariance–three groups combined) | 2242.56 | 711 | < .001 | 3.15 | .041 [.039, .043] | .89 | .06 | 453 | 2764.56 | [.14, .80] |
| M6-factor–Metric invariance (Baseline vs. Measurement) | 2309.88 | 747 | < .001 | 3.09 | .040 [.038, .042] | .89 | .07 | 461 | 2759.88 | [.24, .80] |
| M6-factor–Structural covariances (Measurement vs. Structural) | 2573.09 | 837 | < .001 | 3.07 | .040 [.038, .042] | .88 | .08 | 461 | 2843.09 | [.26, .82] |
| *Third step–alternative models* | | | | | | | | | | |
| M6-factor–alternative model with invariant items unconstrained* | 2268.93 | 735 | < .001 | 3.09 | .040 [.038, .042] | .89 | .06 | 462 | 2742.93 | [.14, .80] |
| M6-factor–alternative model with invariant weights and covariances unconstrained** | 2333.98 | 765 | < .001 | 3.05 | .040 [.038, .042] | .89 | .07 | 466 | 2747.98 | [.14, .81] |

Note.

*The weights for items 12, 15, 17, 18, 20 and 22 were unconstrained as they were found invariant.

**The covariances between autonomy satisfaction and autonomy frustration, autonomy satisfaction and relatedness satisfaction, autonomy frustration and relatedness frustration, relatedness satisfaction and relatedness frustration, autonomy frustration and competence satisfaction, competence frustration and autonomy frustration were unconstrained due to an identified lack of invariance.

for structural covariances were non-positive, we decided to investigate the potential sources of lack of invariance at the level of measurement weights as well as at the level of structural covariances by calculating critical ratios for pairwise comparisons between model parameters in order to find out which parameters were not equivalent. There are several ways of identifying noninvariant items as mentioned by Putnick and Bornstein [56]. Since the problem with nonvariant items was a marginal one in the analyzed case, we decided to use the simplest option for partial-invariance testing, which is a critical ratio analysis delivered by AMOS. The results of the analysis, which are presented in Table 4, showed a lack of equivalence for the following items: item no 18 > autonomy frustration, $\beta_{G1} = .14^*$, $\beta_{G2} = .35^{***}$, $\beta_{G3} = .26^{***}$ with significant pairwise differences between group 1 (G1) and group 2 (G2), $z = 2.66$, $p < .05$, and G2 and group 3 (G3), $z = -1.97$, $p < .05$; item no 15 > autonomy frustration, $\beta_{G1} = .61^{***}$, $\beta_{G2} = .63^{***}$, $\beta_{G3} = .49^{***}$ with a significant difference for G1 and G3, $z = -2.16$, $p < .05$, as well for the G2 and G3, $z = -2.92$, $p < .05$; ***; item 12 > relatedness satisfaction, $\beta_{G1} = .80^{***}$, $\beta_{G2} =$

.82***, $\beta_{G3}$ = .84*** with a significant difference between G1 and G2, $z$ = -2.21, $p$ < .05 and G2 and G3, $z$ = 2.71, $p$ < .05; item no 22 > relatedness frustration, $\beta_{G1}$ = .66***, $\beta_{G2}$ = .72***, $\beta_{G3}$ = .67*** with difference for G1 and G2, $z$ = 2.55, $p$ < .05, and G2 and G3, $z$ = -2.06, $p$ < .05; item no 20 > relatedness frustration, $\beta_{G1}$ = .75***, $\beta_{G2}$ = .78***, $\beta_{G3}$ = .75***, with a significant difference between the G1 and 2, $z$ = 2.06, $p$ < .05; item no 17 > competence frustration, $\beta_{G1}$ = .69***, $\beta_{G2}$ = .77***, $\beta_{G3}$ = .76*** with a significant differences for G1 and G2, $z$ = 2.56, $p$ < .05, as well for the G1 and 3, $z$ = 2.24, $p$ < .05. Summarizing, lack of invariance was related only to six items. No differences for the remaining paths were found, so the majority of paths were invariant. According to some researchers' claims, at least half of the items should be invariant [57] or the majority of items on a factor should show equivalence [51] in order to conclude invariance. Thus, we are confident that the metric invariance of the tool is supported by the data.

For the structural covariances, the significant differences are presented in the lower part of Table 4. Based on the critical ratios expressed in $z$-values, it could be stated that the links between autonomy satisfaction and autonomy frustration, as well for the autonomy frustration and competence satisfaction, were less strongly negative in G1 in comparison to G2 and G3. The relationships between autonomy satisfaction and relatedness satisfaction, as well for autonomy frustration and relatedness frustration, were less strongly positive in G1 in comparison to G2 and G3. Furthermore, in G1 there was a less strongly negative relationship between autonomy frustration and relatedness satisfaction, as well as for relatedness satisfaction and relatedness frustration, and for autonomy satisfaction and relatedness frustration, in comparison to G2. The relationships between relatedness satisfaction and competence satisfaction, as well autonomy satisfaction and competence satisfaction, were both significantly less strong in G1 in comparison to G2. The link between relatedness frustration and competence satisfaction was slightly but still significantly more negative in G1 in comparison to G2. Additionally, the link between competence frustration and autonomy frustration was significantly more positive in G3 in comparison to G1. Also, the relationship between relatedness satisfaction and relatedness frustration was more strongly negative in G2 in comparison to G3. Summarizing, most of the significant differences occurred for the comparison between G1 and G2. The findings related to the invariance analysis are addressed in the discussion.

As a final step, we reran our baseline model with some measurement weights and covariances unconstrained (those which were found invariant) in order to check whether that would improve our models. It turned out that a model with the weights for items 12, 15, 17, 18, 20 and 22 did not significantly differ in comparison to our baseline model ($\Delta\chi^2(24)$ = 26.37, $p$ < .05). This could be treated as another indication that invariance for measurement weights (based on a $\chi^2$ test) was not really a serious problem. However, a model with measurement weights and covariances constrained (those identified as invariant) showed a better fit in comparison to a model with only measurement weights constrained ($\Delta\chi^2(30)$ = 65.05, $p$ < .001), as well as in comparison to the baseline model ($\Delta\chi^2(54)$ = 91.42, $p$ < .01). This shows that invariance related to the structural covariances was more of an issue than the one related to measurement weights.

Concluding, the six-factor structure was confirmed; however, it has to be noted that autonomy satisfaction and competence satisfaction, as well as competence frustration and relatedness frustration are not very well differentiated. Since the six-factor models (overall model and three models for occupational niches separately) were all characterized by a good fit, we recommend that the scale be used in its original form, even though we are aware of some discriminative issues and lack of invariance for several covariances. As discussed later, some of these discrepancies might be related to specificity of examined occupational groups.

**Table 5. Correlations between basic psychological needs, self-efficacy, engagement, job crafting, stress, exhaustion.**

|  | 1 | 2 | 3 | 4 | 5 | 6 | 7 | 8 | 9 | 10 |
|---|---|---|---|---|---|---|---|---|---|---|
| *Satisfaction* |  |  |  |  |  |  |  |  |  |  |
| 1. Autonomy | - |  |  |  |  |  |  |  |  |  |
| 2. Relatedness | .61** | - |  |  |  |  |  |  |  |  |
| 3. Competence | .71** | .66** | - |  |  |  |  |  |  |  |
| *Frustration* |  |  |  |  |  |  |  |  |  |  |
| 4. Autonomy | -.28** | -.28** | -.22** | - |  |  |  |  |  |  |
| 5. Relatedness | -.35** | -.49** | -.49** | .54** | - |  |  |  |  |  |
| 6. Competence | -.36** | -.42** | -.57** | .52** | .74** | - |  |  |  |  |
| 7. Self-efficacy | .55** | .50** | .60** | -.22** | -.32** | -.43** | - |  |  |  |
| 8. Engagement | .56** | .45** | .48** | -.23** | -.19** | -.27** | .60** | - |  |  |
| 9. Job Crafting | .33** | .23** | .18** | -.05 | .11** | .05 | .33** | .44** | - |  |
| 10. Stress | -.29** | -.22** | -.24** | .17** | .11** | .12** | -.28** | -.32** | -.27** | - |
| 11. Exhaustion (burnout) | -.44** | -.36** | -.39** | .50** | .39** | .40** | -.40** | -.40** | -.14** | .44** |

*Note.* N = 1315

*p < .001.

## Criterion validity

Thus far, we have focused on the internal structure of the Polish BPNSFS-Work Domain. In this analysis we examined the convergent validity of the scale using the multitrait-multimethod matrix [58]. The method measures criterion validity by comparing how the constructs from a particular scale relate to similar and different constructs measured by other scales.

Satisfaction of the basic psychological needs correlated positively and moderately with self-efficacy, work engagement and job crafting; and frustration of the needs correlated negatively with those same constructs. Frustration of the basic psychological needs correlated positively with stress and exhaustion, and satisfaction of the basic psychological needs correlated negatively with these constructs (see Table 5). These results corroborate theoretical assumptions and empirical studies [14] and support the criterion validity of the Polish BPNSFS-Work Domain: it does relate to theoretically relevant constructs, but at the same time, the relatively modest relationship between the variables indicates that the constructs are sufficiently distinct.

## Discussion

The purpose of the present study was to adapt and validate the Polish BPNSFS in Work Domain. The scale is a measurement tool which allows the assessment of the degree to which people feel satisfaction (or frustration) of three basic psychological needs from SDT in a professional environment. The three basic psychological needs are generally considered universal across cultures; this is supported by this study, which extends the examination of the basic needs to Polish culture. Moreover, having a validated measurement scale in Polish is crucial to boost new research based on SDT in this community of researchers interested in motivational processes. Having a scale allows us to conduct comparative studies across domains, occupations and groups, and is also useful for meta-analysis purposes. What is more, as need frustration has gained increased research interest around the world, the scale can be used to investigate the dark path of the SDT model [11, 13]. With sound measurements, research efforts in this aspect can be undertaken. Moreover, knowing the benefits of basic psychological need satisfaction and drawbacks of their frustration may contribute to development of new

management styles that are more beneficial for organizations and their workers. These will take into account autonomy, relatedness and competence as key factors.

The findings of the conducted validation study suggest that the Polish scale has robust psychometric features. In line with theory and previous findings [14, 33] the polish BPNSFS-Work Domain proved to have a six-dimensional structure consisting of autonomy satisfaction, autonomy frustration, competence satisfaction, competence frustration, relatedness satisfaction and relatedness frustration. The subscales yielded satisfactory to high internal consistency measured with Cronbach's $\alpha$ ranging from .68 to .82 and McDonalds $\omega$ ranging from .68 to .82. However, in the Polish version of the scale, items 1 (*At work, I feel a sense of choice and freedom in the things I undertake*) and 18 (*My daily activities at work feel like a chain of obligations*) did not achieve high standardized regression weights in the CFA analysis —0.47 and 0.24 respectively. In the general (i.e., not work-related) version of the scale [14] and in its Polish adaptation [31], these two equivalent items yielded a good level of regression weights. We believe that lower coefficients in the present adaptation are related to the characteristics of a work environment, where, in comparison to life 'in general', people feel less freedom and often have a fixed set of obligations or duties.

The six-dimensional structure was also recreated separately for each of the three occupational groups which took part in the study—healthcare workers, education staff and service activities workers. However, although the overall structure of the scale was the same for the three groups, the models were only partially invariant. We believe that these variations are due to the different characteristics of work, tasks and obligations of the three studied groups. For these groups, the subscales of the Polish BPNSFS-Work Domain are composed of the same items, but some of the item-subscale associations' strengths differ. Item 15 (*I feel pressured to do too many things on my job*), which belongs to the autonomy frustration subscale, may be a good example of these between-group differences. For the groups of healthcare workers and educational staff, this item was significantly more strongly related with frustration of autonomy compared to the group of service activity workers. Indeed, the 'pressure to do many things at work' is probably affecting the overall autonomy frustration at work to a greater extent for a doctor or a nurse (healthcare workers), whose job is to care about people's health and save their lives, or a teacher (educational staff), who works with groups and has to be vigilant to many educational needs or requests of his or her students, than for an office worker (service activity worker), who deals with clients to solve mostly administrative matters. Another example could be item 17 (*I feel insecure about my abilities on my job*), which belongs to the competence frustration subscale. Here, for the healthcare worker group, this item was less strongly connected with the competence frustration scale than for the groups of education staff and service activity workers. For the groups of healthcare workers (compared with other investigated groups), this item is probably affecting the overall competence frustration to a lesser extent, as in this kind of profession it is necessary to be secure about one's skills in order to use them with confidence to safeguard the health and lives of patients. We believe that the lack of invariance between some of the covariances is also a natural consequence of the differences between the studied groups. The correlation analysis also discovered that autonomy satisfaction and competence satisfaction, as well as competence frustration and relatedness frustration, are not very well differentiated.

The adapted scale also proved its validity by associations with constructs connected with both positive aspects of work and its 'dark side'. The validity assessment was similar to the validation conducted by Olafsen for Norwegian and English versions of the scale. In Polish adaptation the criterion validity was assessed on the basis of associations with the level of stress, emotional exhaustion (burnout), self-efficacy, job crafting and work engagement. The author of the English and Norwegian validation checked for associations with managerial need

support, motivation (different forms), vigor, emotional exhaustion (burnout), affective commitment and turnover intention. Both in Polish and in Norwegian and English validations it was expected that satisfaction of needs will be positively associated with positive work phenomena (motivation, self-efficacy) and negatively with negative work phenomena (burnout, stress), and with the frustration of needs the relations will be inverted. In both adaptations the results confirmed the researchers' assumptions [34].

The results confirm that constructs such as engagement, job crafting and self-efficacy were positively correlated with satisfaction of the need for autonomy, competence and relatedness, and constructs such as exhaustion (burnout) and stress were positively correlated with frustration of the aforementioned needs. The correlations, however, were moderate. This implies that the constructs of satisfaction and frustration of the basic psychological needs are related to but not interchangeable with the concept of self-efficacy, job-crafting and engagement, as well as stress and burnout. These correlations with both positive and negative organizational phenomena, besides proving the validity of the scale, may be important for organizations for practical reasons. Creating organizational cultures and using managerial styles that satisfy basic psychological needs and prevent workers from being frustrated translates into tangible benefits for employees and organisations [59–61].

## Limitations of the study and directions for further research

Some limitations should be kept in mind when interpreting the results for the present study. Firstly, our research was based on cross-sectional data, meaning that the reliability of the BPNSF was measured by means of a method based on the analysis of the statistical properties of the test items (i.e., internal consistency analysis). In the future, it would be worth conducting longitudinal studies that enable the checking of the instrument's reliability with the method based on the estimation of absolute stability (i.e., the test-retest method), showing to what extent the test results are independent of random factors related to the test person or the test situation and how constant they are over time. Moreover, the research sample is not representative for a general work population. The study was conducted in three occupational groups in which the basic feature of work is direct contact with another person (patient, student or client). Therefore, the research results should not be generalized to other labor market sectors. Further research on satisfaction and frustration of needs in other professional groups is needed. Another potential issue is the disproportionate distribution of the research sample in terms of gender. More than 70% of the respondents were women, so generalization of the results to the male population should be made with care. A further limitation of the study is a reliance on self-report variables measured from the same source. When measuring several variables with the self-report method, there is a risk of occurrence of the common bias method, consisting in artificially inflating the correlation coefficient between the examined variables. Finally, as in any study using self-report measures, the results might be influenced by the participants' acquiescence and need for social desirability. In further research it would be advisable to use some objective measures together with self-report ones. It is worth noting that currently two measures of Basic Psychological Needs are available to study in the Polish language, the general version [31] and the current scale which is intended to measure the needs in work environment.

## Supporting information

**S1 Data.**
(SAV)

**S1 File.**
(PDF)

## Author Contributions

**Conceptualization:** Michał Szulawski, Łukasz Baka, Anja H. Olafsen.

**Data curation:** Michał Szulawski, Łukasz Baka, Monika Prusik.

**Formal analysis:** Michał Szulawski, Łukasz Baka, Monika Prusik.

**Funding acquisition:** Michał Szulawski, Łukasz Baka.

**Investigation:** Michał Szulawski, Łukasz Baka, Monika Prusik.

**Methodology:** Michał Szulawski, Łukasz Baka, Monika Prusik.

**Project administration:** Michał Szulawski, Łukasz Baka.

**Resources:** Łukasz Baka.

**Supervision:** Michał Szulawski, Łukasz Baka, Anja H. Olafsen.

**Validation:** Michał Szulawski, Łukasz Baka, Anja H. Olafsen.

**Visualization:** Michał Szulawski, Łukasz Baka.

**Writing – original draft:** Michał Szulawski, Łukasz Baka.

**Writing – review & editing:** Michał Szulawski, Łukasz Baka, Anja H. Olafsen.

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
