## [Decision Letter · Decision Letter 0]

30 Jun 2021

PONE-D-21-10267

The Basic Psychological Needs Satisfaction and Frustration Scale at Work: A Validation in the Polish Language

PLOS ONE

Dear Dr. Szulawski,

Thank you for submitting your manuscript to PLOS ONE. After careful consideration, we feel that it has merit but does not fully meet PLOS ONE’s publication criteria as it currently stands. Therefore, we invite you to submit a revised version of the manuscript that addresses the points raised during the review process.

We look forward to receiving your revised manuscript.

Kind regards,

Paola Gremigni, Ph.D.

Academic Editor

PLOS ONE

Additional Editor Comments (if provided):

Reviewers' comments:

Reviewer's Responses to Questions

**Comments to the Author**

1. Is the manuscript technically sound, and do the data support the conclusions?

Reviewer #1: Yes

Reviewer #2: Yes

2. Has the statistical analysis been performed appropriately and rigorously? 

Reviewer #1: I Don't Know

Reviewer #2: Yes

3. Have the authors made all data underlying the findings in their manuscript fully available?

Reviewer #1: Yes

Reviewer #2: Yes

4. Is the manuscript presented in an intelligible fashion and written in standard English?

Reviewer #1: Yes

Reviewer #2: Yes

5. Review Comments to the Author

Reviewer #1: This paper seems to be a piece of valuable work to provide a Polish language validated BPNSFS to future researchers. The authors also acknowledged limitations of this study.

As mentioned in the paper that the scale has been validated in Polish by Kuzma et al (2020), is it possible to add a few sentences on how is this validation different to theirs or the further contributions?

I understand that this BPNSFS has 6 dimensions like previous findings, is there any comparisons between the criterion validity of this scale with previous findings?

Reviewer #2: In their manuscript, the authors validate the BPNSF scale within a Polish context. I enjoyed the use of large samples and the strong statistical background. My only larger concern centers around missing a replicable code to rerun the analysis (see comment 1).

I structured my review along the following lines: General comments and minor comments. Each section is structured by perceived importance of the comment. Parts I copy-pasted from the manuscript are in quotation marks.

General comments

1.Some parts of the results section were not 100% clear to me. How exactly was invariance established? When I think of measurement invariance I think of multilevel SEMs (Fischer & Karl, 2019). The approach you are using seems sounds but for less experienced readers an additional explanation (+ a reference at the beginning of the invariance section) that explains the approach or publishing the analysis code (maybe just the AmostOutput?) next to the dataset could be helpful?

2.Related to the above point I was wondering whether a cross-cultural sample would be needed to really establish measurement invariance? At the moment it seems that we can conclude that asking different samples within Poland will produce solid scales across those samples. However, can we compare the findings from Poland to other countries? Maybe this is not the goal of the study though :)

3.I was wondering in which language the questionnaires were distributed. I imagine the BPNSFS was presented in Polish. Were the other items also presented in Polish? Are they validated in that language? If not, would that need to be discussed in the limitations?

4.Within the first paragraph, I was wondering what low-quality vs. high-quality motivation stands for? What Vansteenkiste and colleagues (2009) would term good vs. poor quality motivation groups?

5.Very nice literature review on need frustrations within the organizational context. If the authors would like to broaden their literature review to other areas of research on need frustration some additional references could give the paper a broader audience e.g. (Leander et al., 2019, 2020; Stollberg et al., 2015; Williams, 2009)

6.Nice limitations; great picking up on the gender distribution. Does the dropout (and completion rates leading to them) also need a short discussion?

7.I understand the reason behind summing up the items (Table 1); as there has been some recent debate around obtaining multiple-item scale scores through summing (McNeish & Wolf, 2020), alternative strategies could have been used as well. I am not sure though whether the authors need to engage with this comment; just a thought :)

8.Unfortunately, some of the tables were cut off. Maybe an explanation underneath could help the reader understand the different concepts (e.g. in Table 1 what does the s stand for?)

Minor comments

1.On page 16 it could be handy to refer to the exact number of cases that were excluded? Six I believe?

2.Some sentences could be streamlined (e.g. in the abstract (p. 9) “has been validated” could be “was validated” as the process of validation is now complete)

3.There is still the one or other typo; maybe a fresh pair of eyes could help finding those

a.(e.g. (p. 10) “yet he or she may not feel [incompetence or failure regarding their work]”

b.(p. 11) …, the purpose of this project was to validate the BSNSFS-Work Domain [in Poland].

References

Fischer, R., & Karl, J. A. (2019). A primer to (cross-cultural) multi-group invariance testing possibilities in R. Frontiers in Psychology, 10(JULY), 1–18. https://doi.org/10.3389/fpsyg.2019.01507

Leander, N. P., Agostini, M., Stroebe, W., Kreienkamp, J., Spears, R., Kuppens, T., Van Zomeren, M., Otten, S., & Kruglanski, A. W. (2020). Frustration-affirmation? Thwarted goals motivate compliance with social norms for violence and nonviolence. Journal of Personality and Social Psychology. https://doi.org/10.1037/pspa0000190

Leander, N. P., Stroebe, W., Kreienkamp, J., Agostini, M., Gordijn, E., & Kruglanski, A. W. (2019). Mass shootings and the salience of guns as means of compensation for thwarted goals. Journal of Personality and Social Psychology, 116(5), 704–723. https://doi.org/10.1037/pspa0000150

McNeish, D., & Wolf, M. G. (2020). Thinking twice about sum scores. Behavior Research Methods, 52(6), 2287–2305. https://doi.org/10.3758/s13428-020-01398-0

Stollberg, J., Fritsche, I., & Bäcker, A. (2015). Striving for group agency: threat to personal control increases the attractiveness of agentic groups. Frontiers in Psychology, 6(May), 1–13. https://doi.org/10.3389/fpsyg.2015.00649

Vansteenkiste, M., Sierens, E., Soenens, B., Luyckx, K., & Lens, W. (2009). Motivational Profiles From a Self-Determination Perspective: The Quality of Motivation Matters. Journal of Educational Psychology, 101(3), 671–688. https://doi.org/10.1037/a0015083

Williams, K. D. (2009). Ostracism: A temporal need-threat model. In M. P. Zanna (Ed.), Advances in experimental social psychology, Vol 41. (pp. 275–314). Elsevier Academic Press. https://doi.org/10.1016/S0065-2601(08)00406-1

6. PLOS authors have the option to publish the peer review history of their article (what does this mean?). If published, this will include your full peer review and any attached files.

Reviewer #1: No

Reviewer #2: **Yes: **Maximilian Agostini

---

## [Author Response · Author response to Decision Letter 0]

16 Jul 2021

Dear Editor, 

we addressed additional requirements pointed out in the letter. We addressed: style requirements, added grant information (BSTP 40/20-I) and the Ethics Committee name with the number of agreement (The Maria Grzegorzewska University - Ethics Committee Approval 2020/05 - BSTP 40/20-I). I hope you find these information sufficient, I case of any need please contact the authors for further information. The response to reviewers was attached in additional file.

Kind regards,

Michał Szulawski

---

## [Editor Report · Decision Letter 1]

24 Sep 2021

PONE-D-21-10267R1The Basic Psychological Needs Satisfaction and Frustration Scale at Work: A Validation in the Polish LanguagePLOS ONE

Dear Dr. Szulawski,

Thank you for submitting your manuscript to PLOS ONE. After careful consideration, we feel that it has merit but does not fully meet PLOS ONE’s publication criteria as it currently stands. Therefore, we invite you to submit a revised version of the manuscript that addresses the points raised during the review process. The point-by-point responses you gave to Reviewers are, in my opinion,  competent and exhausive; however, many of these  responses seem not to have been implemented within the manuscript. Can you synthetize and add them? I believe they can answer  other readers who may  potentially raise the same doubts or concerns. 

We look forward to receiving your revised manuscript.

Kind regards,

Prof. Paola Gremigni, Ph.D.

Academic Editor

PLOS ONE
---

## [Author Response · Author response to Decision Letter 1]

8 Oct 2021

Dear Editor,

we have implemented the suggested changes both in response to reviews letter, and in the manuscript. We are sending the revised version.

Kind regards,

Michał Szulawski

---

## [Editor Report · Decision Letter 2]

11 Oct 2021

The Basic Psychological Needs Satisfaction and Frustration Scale at Work: A Validation in the Polish Language

PONE-D-21-10267R2

Dear Dr. Szulawski,

We’re pleased to inform you that your manuscript has been judged scientifically suitable for publication and will be formally accepted for publication once it meets all outstanding technical requirements.

Kind regards,

Paola Gremigni, Ph.D.

Academic Editor

PLOS ONE

Additional Editor Comments (optional):

I only observe that I agree with the authors that invariance analysis may give useful information on the structural stability of a measure  by comparing it across different subpopulations of the same country, besides cross-cultural  comparisons.

---

## [Editor Report · Acceptance letter]

27 Oct 2021

PONE-D-21-10267R2 

The Basic Psychological Needs Satisfaction and Frustration Scale at Work: A Validation in the Polish Language 

Dear Dr. Szulawski:

I'm pleased to inform you that your manuscript has been deemed suitable for publication in PLOS ONE. Congratulations! Your manuscript is now with our production department. 

Kind regards, 

on behalf of

Prof. Paola Gremigni 

Academic Editor

PLOS ONE